# Exercise therapy to improve cervical proprioception in individuals with asymptomatic forward head posture: A systematic review of randomized controlled trials

Ali Asghar Norasteh[1], Kimia Karimi[2], Shabnam Faraji[3], Fereshteh Ejlali[4], Mohammad Alghosi[2,5], Mohammad Alimoradi[6]*, Hamed Zarei[2]

**1** Department of Physiotherapy, Faculty of Medicine, Guilan University of Medical Sciences, Rasht, Iran, **2** Department of Sports Injury and Corrective Exercise, Faculty of Physical Education & Sport Sciences, University of Guilan, Rasht, Iran, **3** Department of Sport Injuries and Biomechanics, Faculty of Sport Sciences and Health, University of Tehran, Tehran, Iran, **4** Department of Physical Education and Exercise Science, Islamic Azad University, Central Tehran Branch (Velayat Campus), Tehran, Iran, **5** Department of Physical Education, Technical and Vocational University (TVU), Tehran, Iran, **6** Department of Sports Injuries and Corrective Exercises, Faculty of Sports Sciences, Shahid Bahonar University of Kerman, Kerman, Iran

* malimoradi@sport.uk.ac.ir

## Abstract

### Background

Forward head posture (FHP) is a common musculoskeletal condition associated with impaired cervical proprioception, which compromises postural control and neuromuscular function. Exercise-based interventions have been proposed to address proprioceptive deficits in FHP, but their effectiveness remains unclear. This systematic review aimed to evaluate the impact of exercise programs on cervical proprioception in individuals with FHP.

### Methods

A systematic search of PubMed, Scopus, and Web of Science was conducted up to April 20, 2025. Randomized controlled trials (RCTs) assessing exercise interventions on cervical proprioception in FHP were included. Risk of bias was assessed using the RoB-2 tool. Due to heterogeneity in outcome measures, a narrative synthesis was conducted, supported by effect size (ES) calculations.

### Results

Nine RCTs involving 367 participants were included. Interventions ranged from cervical stabilization exercises (most common) to whole-body vibration, backward walking, and muscle energy techniques. ESs varied from trivial to nearly perfect, with trivial-to-very-large improvements observed in joint position sense and joint position error.

**Data availability statement:** All relevant data are within the manuscript and its Supporting Information files.

**Funding:** The author(s) received no specific funding for this work.

**Competing interests:** The authors have declared that no competing interests exist.

While cervical stabilization exercises demonstrated positive outcomes in rotation and flexion tasks across some studies, the limited number and heterogeneity of studies on alternative interventions precluded a definitive comparison of consistency or effectiveness. Risk of bias was generally rated as "some concerns" due to lack of blinding and variability in outcome measures.

## Conclusion

Current evidence suggests cervical stabilization exercises, the most studied intervention for FHP, may improve cervical proprioception. However, methodological heterogeneity and diagnostic inconsistencies, such as variations in craniovertebral angle thresholds used to define FHP, limit the ability to draw definitive conclusions. Future studies should standardize diagnostic criteria, outcome assessments, and investigate long-term effects across diverse populations.

## Introduction

Proprioception, the body's ability to sense its position, movement, and spatial orientation, is fundamental for motor control, postural stability, and injury prevention [1–4]. In the cervical spine, proprioception plays a critical role in coordinating head and neck movements, maintaining balance, and stabilizing the head during dynamic activities [5,6]. The cervical region is densely populated with muscle spindles and mechanoreceptors, making it highly reliant on accurate proprioceptive input for proper function [7]. Disruptions in cervical proprioception can lead to impaired joint position sense (JPS), increased joint position error (JPE), and a higher risk of musculoskeletal dysfunction, reduced threshold to detect passive motion, and a higher risk of musculoskeletal dysfunction [8,9].

One of the most common postural deviations affecting cervical proprioception is forward head posture (FHP), characterized by anterior displacement of the head relative to the shoulders and an decreased craniovertebral angle (CVA) (8). FHP is prevalent across various populations, particularly among individuals with sedentary lifestyles, prolonged screen use, or occupations requiring sustained forward head positions [10]. The condition is associated with multiple adverse effects, including neck pain, muscle fatigue, reduced respiratory efficiency, and reduced quality of life [11–13]. Importantly, FHP has been linked to reduced cervical proprioception, as the altered alignment may disrupt afferent feedback from cervical muscle spindles and joint mechanoreceptors [14,15]. Studies suggest that individuals with FHP exhibit greater JPE compared to those with normal posture, indicating reduced proprioceptive acuity [16,17].

Despite growing recognition of the relationship between FHP and proprioceptive deficits, there remains a lack of consensus on diagnostic criteria for FHP. Various studies have proposed different CVA thresholds (e.g., ≤ 53°, < 50°, < 49°, or <48°), leading to inconsistencies in study populations and intervention outcomes [18–21]. Additionally, while training interventions [20,22], such as cervical stabilization

exercises and postural correction,have shown promise in improving cervical proprioception, the optimal exercise prescription (frequency, intensity, type, and duration) remains unclear. Some studies suggest that proprioceptive training yields greater improvements than conventional strength training, while others highlight the benefits of combined approaches [23]. Furthermore, the mechanisms by which different exercises enhance proprioception in FHP, whether through neuromuscular re-education, enhanced muscle spindle sensitivity, or cortical adaptation, require further exploration.

This systematic review aims to synthesize existing evidence on the effectiveness of interventions in improving cervical proprioception in individuals with FHP. By evaluating the methodologies, outcomes, and limitations of current studies, this review seeks to clarify which interventions are most effective and identify gaps for future research. The findings may inform clinical practice by guiding evidence-based rehabilitation strategies for individuals with FHP-related proprioceptive impairments.

## Methods

### Protocol and registration

The systematic review follows the Preferred Reporting Items for Systematic Reviews and Meta-Analyses (PRISMA) guideline [24] and was registered in the International Prospective Register of Systematic Reviews (PROSPERO) with registration number CRD42023488327.

### Eligibility criteria

Following the search phase, AAN and KK independently conducted a thorough review of all retrieved titles and abstracts. The systematic review's study inclusion process was carefully structured using the PICOS (Population, Intervention, Comparison, Outcome, and Study Design) criteria [25], ensuring a methodical and transparent approach to selecting relevant studies (Table 1).

### Search strategy

The search included the electronic databases Web of Science, PubMed, and Scopus, from inception to April 20, 2025, with two authors (KK and MALG) searching independently, with discrepancies resolved through discussion and, if needed, the opinion of a third author. The search strategy used specific Mesh terms and text words/phrases, combined using

**Table 1. Selection criteria for studies.**

|  | Inclusion criteria | Exclusion criteria |
|---|---|---|
| **Population** | • Individuals with FHP without pain<br>• Individuals with ≥ 18 years old<br>• Any sex | Individuals with cervical pathology |
| **Intervention** | Any movement-based intervention | More than one intervention was compared at the same time |
| **Comparison** | Control group without exercise, sham exercise group, or condition for comparison | Not applicable |
| **Outcomes** | Investigate the effect of the intervention on joint position error related to cervical proprioception (e.g., joint position error) | No related outcome |
| **Study Design** | RCTs and non-RCTs | Single-group intervention; Case studies; Reviews. |

**Abbreviations:** RCTs, randomized controlled trials; non-RCTs, non-randomized controlled trials.

Boolean operators (e.g.,(exercise OR "physical activity" OR training) AND (cervical OR neck OR head) AND ("propriocep-tion" OR "sense of equilibrium" OR "equilibrium sense" OR "position sense" OR "posture sense" OR "sense of position" OR "kinesthesis" OR "joint position sense" OR "sense of resistance")). The specific search strategy for each database is presented in Table 2. This review had no restrictions on language, and Google Translate was used to interpret studies not published in English. Additionally, the reference lists of the included studies were screened, and a grey literature search was conducted using Google Scholar to identify further relevant studies, including conference proceedings, dissertations, clinical trial registries, and preprint servers (e.g., medRxiv, bioRxiv). The retrieved studies were organized using Endnote, and any duplicate entries were removed. The Connected Papers website (https://www.connectedpapers.com/) was used to enhance the search for relevant research.

## Data extraction

Two authors (SF and FE) collected information from retrieved papers, including study details (author, year of publication, loca-tion), study design, sample description (sample size, sex, age and craniovertebral angle), exercise characteristics, experiment group intervention program, control group intervention, cervical proprioception measures, assessment tool, main outcomes, conclusion, and risk of bias (ROBINS-I and RoB-2 overall judgment). If any important information was missing, the corre-sponding authors were contacted via email, with a maximum of three attempts made to obtain the necessary details.

## Quality assessment

The risk of bias in the included randomized controlled trials (RCTs) was independently assessed by two reviewers (MALG and MALI) using the Cochrane Risk of Bias Tool (RoB-2) [26] across five domains: randomization process, deviations from intended interventions, missing outcome data, outcome measurement, and selection of reported results. Each domain was evaluated through signaling questions and judged as "low risk," "some concerns," or "high risk" of bias Discrepan-cies were resolved through discussion to ensure consistency. The randomization process was scrutinized for sequence generation and allocation concealment, while deviations from interventions examined protocol adherence and intention-to-treat analysis. Missing data were assessed for attrition rates and handling methods, and outcome measurement focused on assessor blinding and objectivity. Selective reporting was verified by comparing pre-specified and published outcomes.

**Table 2. Databases search strategies.**

| Database | Complete search strategy |
|---|---|
| Web of Science | exercise OR "physical activity" OR training (Topic) AND cervical OR neck OR head (Topic) AND "proprioception" OR "vestibular sense" OR "sense of equilibrium" OR "equilibrium sense" OR "labyrinthine sense" OR "position sense" OR "posture sense" OR "sense of position" OR "kinesthesis" OR "joint position sense" OR "sense of resistance" (Topic) |
| PubMed | ((exercise[Title/Abstract] OR "physical activity"[Title/Abstract] OR training[Title/Abstract]) AND (cervical[Title/Abstract] OR neck[Title/Abstract] OR head[Title/Abstract])) AND ("proprioception"[Title/Abstract] OR "vestibular sense"[Title/Abstract] OR "sense of equilibrium"[Title/Abstract] OR "equilibrium sense"[Ti-tle/Abstract] OR "labyrinthine sense"[Title/Abstract] OR "position sense"[Title/Abstract] OR "posture sense"[Title/Abstract] OR "sense of position"[Title/Abstract] OR "kinesthesis"[Title/Abstract] OR "joint position sense"[Title/Abstract] OR "sense of resistance"[Title/Abstract]) |
| Scopus | (TITLE-ABS-KEY (exercise OR "physical activity" OR training) AND TITLE-ABS-KEY (cervical OR neck OR head) AND TITLE-ABS-KEY ("proprioception" OR "vestibular sense" OR "sense of equilibrium" OR "equilibrium sense" OR "labyrinthine sense" OR "position sense" OR "posture sense" OR "sense of position" OR "kinesthesis" OR "joint position sense" OR "sense of resistance")) |

Results were visualized using the Robvis tool to generate traffic light plots, providing a clear representation of bias across studies in accordance with Cochrane guidelines.

## Data synthesis

The study used narrative data synthesis and followed the PRISMA guidelines [24] to ensure thorough and transparent reporting, improving the validity of the results. Due to different outcome measures, a meta-analysis was not possible, so the review used a Synthesis Without Meta-Analysis (SWiM) approach, following the SWiM reporting guideline [27]. To provide specific evidence, effect sizes (ESs) were computed from reported means and standard deviations and classified based on Hopkins' scale as trivial (<0.2), small (0.2–0.6), moderate (0.6–1.2), large (1.2–2.0), very large (2.0–4.0), and nearly perfect (> 4.0) [28]. The ES for each study was computed using the following formula Cohen's $d$:

$$d = \frac{M_1 - M_2}{SD_{pooled}}$$

Where $M_1$ and $M_2$ represent the means of the groups being compared, and $SD_{pooled}$ is the pooled standard deviation, calculated as:

$$SD_{pooled} = \sqrt{\frac{(n_1 - 1) \times SD_1^2 + (n_2 - 1) \times SD_2^2}{n_1 + n_2 - 2}}$$

Where $n_1$ and $n_2$ are the sample sizes and $SD_1$ and $SD_2$ are the standard deviations for each group. Following this method, the computed ESs were classified according to Hopkins' scale to describe the strength of the effects of exercise interventions on cervical proprioception. Additionally, prior to the data merging, the level of agreement between reviewers at each stage of the evaluation process was systematically assessed using Kappa (κ) statistics. The strength of agreement was categorized into distinct levels: poor (κ ≤ 0.20), fair (κ = 0.21–0.40), moderate (κ = 0.41–0.60), substantial (κ = 0.61–0.80), or near-perfect (κ = 0.81–0.99) [29].

## Certainty of evidence assessment

The certainty of the evidence for each primary outcome was evaluated using the GRADE (Grading of Recommendations, Assessment, Development, and Evaluation) approach, as described in the Cochrane Handbook for Systematic Reviews of Interventions. The GRADE framework assesses evidence across five domains: risk of bias, inconsistency, indirectness, imprecision, and publication bias. Outcomes were rated as high, moderate, low, or very low certainty [30]. Since all included studies were randomized controlled trials, the starting level of certainty was high and downgraded as necessary based on the identified limitations.

## Results

### Study identification

Based on the PRISMA guidelines [24], the electronic databases search process initially retrieved 791 articles, but after removing duplicates, 574 (72.6%) studies were left for further screening based on the inclusion and exclusion criteria. The review focused on studies that investigated the effects of therapeutic exercises on cervical proprioception in individuals with forward head posture. A summary of the study progression and the rationale for exclusions at each phase is depicted in a PRISMA diagram (Fig 1).

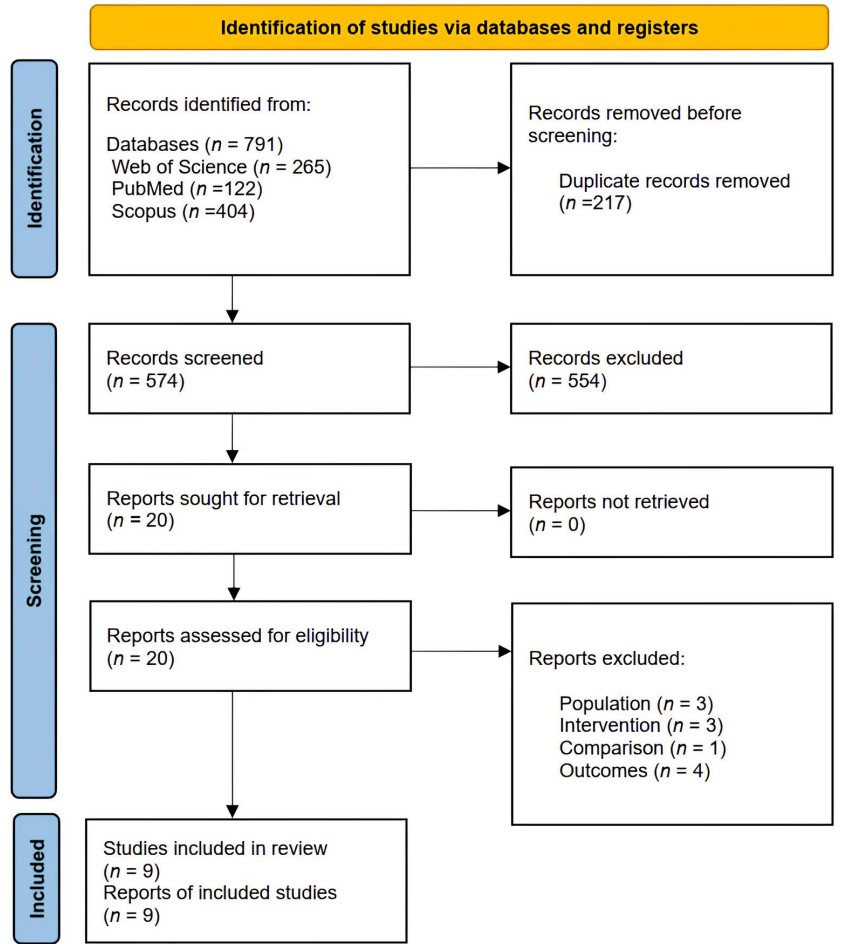

**Fig 1. Preferred Reporting Items for Systematic Reviews and Meta-Analyses (PRISMA) 2020 flow diagram for new systematic reviews, including searches of databases and registers.**

## Descriptive characteristics of the included studies

The nine studies included in this systematic review, published between 2018 and 2024, encompassed 367 participants. Among the 319 participants with reported sex distribution, 102 (32.0%) were male and 217 (68.0%) were female. Studies were conducted across Iran (n=4), Korea (n=2), Thailand (n=1), and India (n=2). All studies utilized RCT designs with sample sizes ranging from 27 to 60 participants, featuring a mean age of 22.6±2.1 years and a predominance of female participants in three studies [21,31,32]. Moreover, one study did not mention the sex of the participants [33]. Considerable variation existed in the diagnostic criteria for FHP, CVA thresholds ranging from <48° to <60° [18–21,31–34] and one study using a broader range of 40°-60° [35], highlighting a key methodological inconsistency across studies. The included studies investigated eight distinct training protocols for improving cervical proprioception in individuals with FHP. Cervical stabilization exercises represented the most common intervention (n=3/9), implemented in Parishan et al. (2021), Goo et al. (2024), and Lee et al. (2022) [18,21,34].Other interventions included scapular stabilization exercise [34], postural correction with muscle co-contraction and stretching [20], whole-body vibration training (WBV) [19], backward walking on a treadmill [32], the National Academy of Sports Medicine corrective exercise protocol [31], cervical retraction combined with

muscle energy techniques [35], and conventional physiotherapy incorporating hot packs, ultrasound therapy, and posture correction exercises [33]. Program durations ranged from single sessions to 8 weeks, with treatment frequencies varying between daily to 3–7 sessions per week. This diversity of approaches reflects the current variability in clinical practice for managing FHP-related proprioceptive deficits. Control conditions consisted of no intervention (n = 3) [20,31,32], sham exercises (n = 1) [19], or were absent (n = 5) [18,21,33–35]. Outcome measures were similarly diverse: five studies assessed JPE using laser pointers or digital inclinometers [18,20,21,32,34], while four evaluated JPS through range-of-motion instruments or repositioning tests [19,31,33,35]. Table 3 provides a summary of the specific characteristics and retrieved data of all nine studies.

## Effect Sizes of Interventions on JPS and JPE

The calculated ESs for interventions targeting cervical proprioception demonstrated clinically meaningful improvements across all studies, though the magnitude varied by intervention type and outcome measure (JPS vs. JPE). For JPS outcomes, four studies showed ES values for flexion changes ranging from small to nearly perfect (ES = 0.33–4.55) [19,31,33,35], while four studies reported ES values for extension changes ranging from trivial to nearly perfect (ES = 0.11–4.52) [19,31,33,35]. Three studies demonstrated moderate to very large ES values for right rotation changes (ES = 1.11–3.59) [31,33,35], and three studies showed large to very large ES values for left rotation changes (ES = 1.98–2.98) [31,33,35]. Additionally, one study found very large ES values for right flexion changes (ES = 2.30) and left flexion changes (ES = 3.13) [33]. For JPE outcomes, two studies reported ES values for flexion changes ranging from trivial to very large (ES = 0.19–3.66) [18,21], while two studies showed ES values for extension changes ranging from trivial to small (ES = 0.12–0.58) [18,21]. For the joint relocation test, two studies found moderate to large ES values (ES = 1.19–1.46) [32,34]. Additionally, one study reported moderate ES values for right rotation changes (ES = 0.86) and left rotation changes (ES = 0.63) [20]. Fig 2 illustrates all the ESs of interventions on JPS and JPE. To improve interpretability, Table 4 summarizes the ESs by intervention type.

## Quality assessment

All nine included RCTs were assessed for risk of bias using the RoB-2 tool (Fig 3), with each study exhibiting "some concerns" in the overall rating. Domain-specific analysis revealed that most studies (6/9) raised concerns in Domain 2 (bias due to deviations from intended interventions) [19–21,33–35], while 6/9 showed issues in Domain 4 (bias in outcome measurement) [18,19,31–34]. Domain 1 (randomization) was rated as low risk in 6 studies [18–21,33,34] and some concerns in 3 [31,32,35]. All studies demonstrated low risk in Domains 3 (missing outcome data) and 5 (selection of reported results) [18–21,31–35]. Inter-rater agreement was perfect (κ = 1.0), confirming consistent bias assessments across reviewers. The findings highlight methodological limitations, particularly in intervention adherence and outcome measurement, across the majority of included trials.

## Certainty of the evidence

The certainty of the evidence was assessed using the GRADE approach. For the outcome of JPS, the certainty of evidence was rated as Low, due to some concerns about inconsistency related to variability in interventions and measurement tools. For JPE, the certainty of evidence was rated as low, primarily due to inconsistency and imprecision caused by small sample sizes and heterogeneous outcome measures. A summary of findings is presented in Table 5.

## Discussion

This systematic review evaluated the effectiveness of exercise-based interventions in improving cervical proprioception among individuals with FHP. The findings suggest that various exercise modalities,including cervical stabilization exercises, postural correction, WBV, and muscle energy techniques,can enhance proprioceptive acuity, as measured by JPS and JPE. However, the heterogeneity in exercise protocols, and outcome measures underscores the need for cautious interpretation and highlights gaps for future research.

**Table 3. Characteristics of the included studies.**

| Study details | Design of study | Sample Description | Exercise characteristics | Control group Intervention | Cervical proprioception measures | Assessment tool | Main outcomes | Conclusion | Risk of bias |
|---|---|---|---|---|---|---|---|---|---|
| Salami et al. (2018) [19] Iran | RCT | N = 30 individuals with FHP Sex = 18 males and 12 females Age = 21.4 ± 2.1 years CVA = < 48º | D = 1 session F = NA I = NA T = NA T = WBV | Holding the same position for the same time without any vibration. | JPS | Cervical range of motion instrument | A significant reduction (P < 0.05) in JPS absolute errors in 2 target angles (adapted head posture and 50% extension ROM) in the vibration group. | A relative improvement in cervical JPS can be achieved when adding additional sensory input from whole body vibration stimulus to the head and neck retraction exercise. | Some concerns |
| Jantoon and Uthaikhup (2020) [20] Thailand | RCT | N = 53 individuals with FHP Sex = 18 males and 35 females Age = 21.8 ± 1.4 years CVA = < 50º | D = 4 weeks F = 7 times a week I = NA T = NA T = Postural correction, muscle co-contraction, and muscle stretching | No exercise | JPE | A laser pointer attached to the head | JPEs (rotation to right and rotation to left) before and after the exercise program were significantly found in the experimental group (P < 0.05). | The 4-week exercise program was effective in improving the joint position sense. | Some concerns |
| Parishan et al. (2021) [21] Iran | RCT | N = 43 individuals with FHP Sex = 43 females Age = 23.8 ± 3.3 years CVA = < 49º | D = 4 weeks F = 2 times a day and 3 times a week I = NA T = NA T = Cervical stabilization exercise | NA | JPE | Digital inclinometer | A significant difference in repositioning error angles was found before and one month after the intervention (P < 0.05). | Stabilization exercise was effective for reducing neck repositioning error angle and improving proprioception in individuals with FHP. | Some concerns |
| Shah et al. (2022) [35] India | RCT | N = 60 individuals with FHP Sex = 33 males and 27 females Age = 23.3 ± 1.5 years CVA = 40°-60° | D = 2 weeks F = 3 times a week I = NA T = NA T = Cervical retraction exercise and muscle energy technique | NA | JPS | Head repositioning accuracy test | The study found a significant reduction in error in cervical JPS within the groups (P < 0.05) and between the groups (P < 0.05) for all physiological movements. | Cervical Retraction as well as Muscle Energy Technique can be used to improve Cervical JPS. However, Cervical Retraction is better as compared to Muscle Energy Technique. | Some concerns |
| Lee et al. (2022) [34] Korea | RCT | N = 27 individuals with FHP Sex = 15 males and 12 females Age = 21.3 ± 1.1 years CVA = < 60º | D = 6 weeks F = 3 times a week I = NA T = 30 minutes T = Cervical stabilization and scapular stabilization exercises | NA | JPE | A laser pointer attached to the head | There was a significant difference after the training programs in head repositioning error (P < 0.05). | Cervical and scapula stabilization exercises were effective to improve neck proprioception. | Some concerns |
| Abdolahzadeh and Daneshmandi, (2023) [31] Iran | RCT | N = 30 individuals with FHP Sex = 30 females Age = 20.2 ± 1.7 years CVA = > 46º | D = 8 weeks F = 3 times a week I = NA T = 30–70 minutes T = NASM's protocol | No exercise | JPS | Head repositioning error test | Cervical JPS in the intervention group had a significant improvement compared to the control group (P ≤ 0.001). | Corrective exercises based on NASM's protocol seem to improve the cervical joint position sense. | Some concerns |
| Goo et al. (2024) [18] Korea | RCT | N = 30 individuals with FHP Sex = 18 males and 12 females Age = 23.3 ± 1.5 years CVA = ≤ 53º | D = 4 weeks F = 4 times a week I = NA T = 30 minutes T = Cervical stabilization exercise | NA | JPE | Digital inclinometer | There was a significant difference after the intervention in extension (P < 0.05). | The cervical stabilization exercise was effective for the proprioception of subjects. | Some concerns |

*(Continued)*

**Table 3.** (Continued)

| Study details | Design of study | Sample Description | Exercise characteristics | Control group Intervention | Cervical proprioception measures | Assessment tool | Main outcomes | Conclusion | Risk of bias |
|---|---|---|---|---|---|---|---|---|---|
| Mahmoudi et al. (2024) [32] Iran | RCT | N = 46 individuals with FHP Sex = 46 females Age = 22.8 ± 2.3 years CVA = > 38º and < 52º | D = 4 weeks F = 4 times a week I = Treadmill speed (2.4 km/h first two weeks then increased to 3.4 km/h) T = 10 minutes T = Backward walking on treadmill | No exercise | JPE | A laser pointer attached to the head | There was a significant difference after the training program in neck proprioception (P = 0.001). | Backward walking enhanced the proprioception in individuals with FHP. | Some concerns |
| Panihar and Joshi (2024) [33] India | RCT | N = 48 individuals with FHP Sex = NA Age = 26.1 ± 4.7 years CVA = < 50º | D = 6 weeks F = 3 times per week I = NA T = NA T = Conventional physiotherapy: hot pack, ultrasound therapy and posture correction exercises | NA | JPS | Head repositioning error test | There was a significant difference after the training program in head repositioning error (P < 0.001). | Conventional treatment improved the proprioception in individuals with FHP. | Some concerns |

**Abbreviations:** CVA, craniovertebral angle; D, duration; F, frequency; FHP, forward head posture; I, intensity; JPE, joint position error; JPS, joint position sense; N, number; NA, not applicable; WBV, whole body vibration; NASM, National Academy of Sports Medicine; RCT, randomized controlled trial; ROM, range of motion; T, time/type.

### Key findings and mechanisms of improvement

The reviewed studies consistently demonstrated that exercise interventions led to clinically meaningful improvements in cervical proprioception. For instance, cervical stabilization exercises showed improvements in JPS and JPE with ESs ranging from 0.19 to 3.66 in flexion task, while joint relocation improvements with a value of 1.40, highlighting substantial proprioceptive gains. These exercises, the most frequently studied intervention (n = 3), were particularly effective in reducing JPE and enhancing JPS [18,21,34]. These exercises target deep cervical flexors and scapular stabilizers, which are often weakened in FHP, thereby restoring neuromuscular control and afferent feedback from muscle spindles and mechanoreceptors [7]. For example, Parishan et al. (2021) reported significant reductions in repositioning errors in flexion (ES = 1.12–3.66) after 4 weeks of stabilization training, suggesting that neuromuscular re-education plays a critical role in proprioceptive recovery [21]. The efficacy of these exercises can be attributed to several key physiological mechanisms:

**Neuromuscular re-education**: Neuromuscular re-education refers to the process of retraining the nervous system to improve the coordination and activation of muscles, particularly those responsible for postural control and joint stability. In the context of FHP, prolonged anterior displacement of the head leads to overactivity of superficial muscles (e.g., sternocleidomastoid and upper trapezius) and underactivity of deep cervical flexors (e.g., longus colli and longus capitis) [36,37]. Stabilization exercises target these deep muscles, promoting their reactivation and restoring optimal muscle activation patterns. This re-education process enhances dynamic joint stability and reduces reliance on superficial muscles, which are prone to fatigue and inefficient for proprioceptive feedback [38].

**Enhanced muscle spindle sensitivity**: The deep cervical flexors are richly innervated with muscle spindles, which play a critical role in proprioceptive acuity [7]. In FHP, prolonged lengthening of these muscles may lead to spindle desensitization [39]. Stabilization exercises, particularly those involving slow, controlled movements with an emphasis on precision (e.g., chin tucks with head nods), may increase spindle sensitivity by restoring optimal muscle length-tension relationships [40].

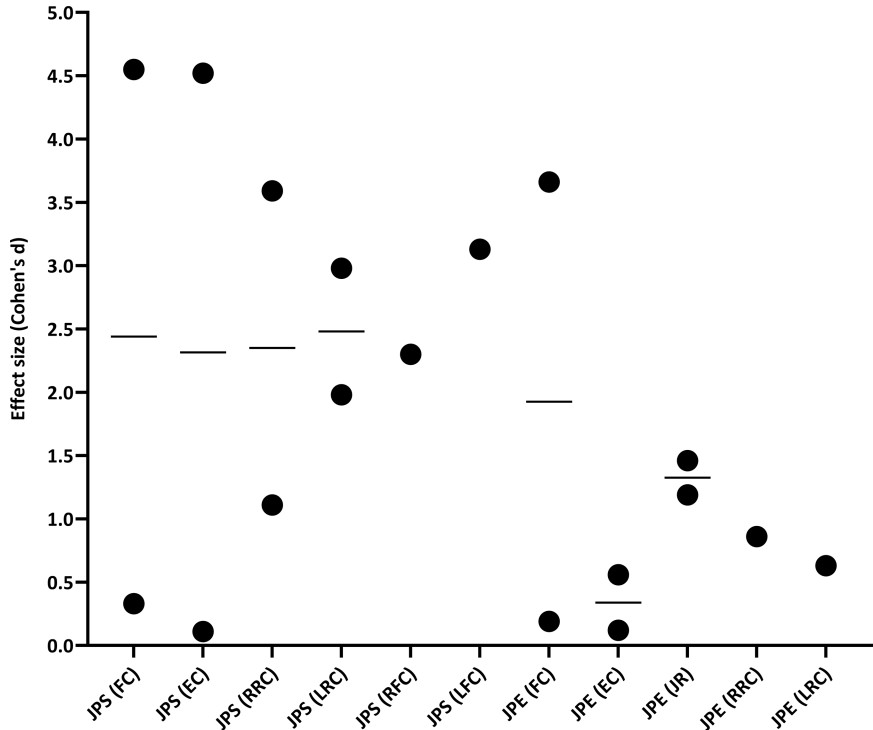

**Fig 2. Effect sizes (Cohen's d) of interventions on JPS and JPE.** Black circles (●) indicate individual effect sizes from each study, including those with the smallest and largest values. Horizontal lines (—) represent the mean effect size within each intervention category. Abbreviations: JPS: joint position sense, JPE: joint position error, FC: flexion changes, EC: extension changes, RRC: right rotation changes, LRC: left rotation changes, RFC: right flexion changes, LFC: left flexion changes, and JR: joint relocation.

## Alternative interventions

Emerging evidence suggests that proprioceptive rehabilitation for FHP may benefit from a diverse range of interventions beyond traditional cervical stabilization exercises. WBV has shown particular promise, with Salami et al. (2018) demonstrating immediate improvements in JPS (ESs: flexion = 0.33, extension = 0.11) following a single session [19]. This acute effect likely stems from WBV's unique ability to stimulate Ia afferent fibers through tonic vibration reflexes, temporarily enhancing muscle spindle sensitivity and facilitating cortical proprioceptive remapping [19,41]. However, the clinical utility of WBV remains uncertain due to the absence of long-term follow-up data and standardized protocols for vibration parameters (frequency, amplitude, duration). Similarly innovative, backward walking has emerged as a potentially valuable intervention, with Mahmoudi et al. (2024) reporting a large effect on joint repositioning accuracy (ES = 1.19) following a 4-week program [32]. This approach appears to work through multiple mechanisms – it challenges the postural control system in novel ways that demand increased reliance on cervical proprioceptive inputs while simultaneously promoting co-activation of deep cervical flexors and scapular retractors [42,43]. Despite these promising results, practical implementation barriers exist, particularly the requirement for specialized equipment and close supervision.

Complementary approaches that incorporate elements of postural re-education or muscle energy techniques have individually shown efficacy in improving cervical JPS and JPE. For instance, Jantoon and Uthaikhup (2020) demonstrated benefits from postural correction combined with muscle co-contraction and stretching (ESs: extension = 0.12, right rotation = 0.86, left rotation = 0.63) in the JPE [20], while Shah et al. (2022) reported improvements using cervical retraction (ESs: flexion = 3.05, extension = 4.52, right rotation = 3.59, left rotation = 2.97) and muscle energy techniques (ESs:

**Table 4. Summary of effect sizes by intervention type.**

| Intervention | Measurement | ES range | Magnitude | Studies |
|---|---|---|---|---|
| Cervical stabilization exercise | Flexion | 0.19–3.66 | Trivial to very large | Parishan et al. and Goo et al. |
| | Joint relocation | 1.40 | large | Lee et al. |
| Scapular stabilization exercise | Joint relocation | 1.46 | large | Lee et al. |
| Postural correction | Extension | 0.12 | Trivial | Jantoon and Uthaikhup |
| | Right rotation | 0.86 | Moderate | |
| | Left rotation | 0.63 | Moderate | |
| WBV | Flexion | 0.33 | Small | Salami et al. |
| | Extension | 0.11 | Trivial | |
| Backward walking on treadmill | Joint relocation | 1.19 | Moderate | Mahmoudi et al. |
| NASM | Flexion | 2.14 | Very large | Abdolahzadeh and Daneshmandi |
| | Extension | 0.87 | Moderate | |
| | Right rotation | 2.02 | Very large | |
| | Left rotation | 2.23 | Very large | |
| Cervical retraction combined with muscle energy techniques | Flexion | 3.05–4.55 | Very large to nearly perfect | Shah et al. |
| | Extension | 1.69–4.52 | Large to nearly perfect | |
| | Right rotation | 1.11–3.59 | Moderate to very large | |
| | Left rotation | 1.98–2.97 | Large to very large | |
| Conventional physiotherapy | Flexion | 1.69 | Large | Panihar and Joshi |
| | Extension | 2.48 | Very large | |
| | Right rotation | 3.44 | Very large | |
| | Left rotation | 2.98 | Very large | |
| | Right flexion | 2.30 | Very large | |
| | Left Flexion | 3.13 | Very large | |

**Abbreviations:** ES, effect size; WBV, whole body vibration; NASM, National Academy of Sports Medicine.

flexion = 4.55, extension = 1.69, right rotation = 1.11, left rotation = 1.98) in the JPS [35]. These distinct but synergistic strategies may offer particular advantages for patients with FHP or myofascial pain by addressing both soft tissue limitations and neuromuscular control deficits.

While these alternative modalities expand the therapeutic options for addressing proprioceptive impairments associated with FHP, their relative efficacy compared to conventional cervical stabilization exercises remains unclear. This uncertainty primarily arises from the limited number of high-quality randomized controlled trials directly comparing these interventions, the heterogeneity in study designs, and the lack of standardized treatment protocols regarding dosage, frequency, and duration. Furthermore, many studies focus on short-term outcomes without sufficient long-term follow-up, hindering the ability to determine sustained clinical benefits.

### Exercise prescription considerations

Analysis of the included studies (Table 3) reveals variability in exercise prescription parameters for improving cervical proprioception in FHP. While cervical stabilization exercises (3–4 sessions/week for 4–6 weeks) emerged as the most studied intervention (ES range: 0.19–3.66) [19,21,34] critical gaps remain in our understanding of optimal dosing. Only one study specified exercise intensity (backward walking at 2.4–3.4 km/h) [32], while others lacked objective intensity measures, and none compared different frequencies or durations head-to-head. Alternative approaches like WBV (single session) and muscle energy techniques (2 weeks) showed promise but require longer-term study [19,35]. Notably, none of the nine

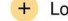

**Fig 3. Risk of bias summary for randomized controlled trials evaluated with RoB-2.**

**Table 5. Summary of findings (GRADE).**

| Outcome | No. of studies | Participants | Effect estimate | Certainty (GRADE) | Reason for downgrade |
|---------|----------------|--------------|-----------------|-------------------|----------------------|
| JPS | 4 RCTs | ~168 | Trivial to nearly perfect improvement (ES: 0.11–4.55) | ●●○○ Low | Inconsistency + imprecision: diverse tools and interventions, small samples |
| JPE | 5 RCTs | ~199 | Trivial to very large improvement (ES: 0.12–3.66) | ●●○○ Low | Inconsistency + imprecision: diverse tools and interventions, small samples |

**Abbreviations:** GRADE, Grading of Recommendations, Assessment, Development, and Evaluation; JPS, joint position sense; JPE, joint position error; RCTs, randomized controlled trials.

**Certainty ratings:** ●●●● High, ●●●○ Moderate, ●●○○ Low, ●●○○○ Very low.

studies compared different frequencies or durations directly rates [18–21,31–35], and only 4/9 specified session duration (10–70 minutes) [18,31,32,34]. This heterogeneity in frequency (daily to 7 × /week), duration (single session to 8 weeks), and unspecified intensities underscores the need for standardized protocols and comparative effectiveness research to establish evidence-based prescription guidelines.

## Limitations and future directions

Across all included studies, none reported adverse effects of exercise programs on proprioception, with 77.77% of measured outcomes demonstrating moderate-to-nearly perfect ES values [20,21,31–35]. This consistency underscores the

clinical relevance of proprioceptive training in managing FHP-related dysfunctions, despite methodological variability. However, the current evidence base for exercise interventions targeting cervical proprioception in FHP has several key limitations that warrant careful consideration. A primary concern is the substantial methodological heterogeneity across studies, particularly regarding diagnostic criteria and outcome measurement. The lack of consensus on FHP classification is particularly problematic, with studies employing varying CVA thresholds ranging from <48° to <60° [18–21,31–34], and one study using an exceptionally broad range of 40–60° [35]. This diagnostic inconsistency has important clinical implications, as research by Titcomb et al. (2024) suggests that even a 5° difference in CVA measurement could lead to misclassification of approximately 20% of cases [10], potentially confounding study results and limiting their generalizability. Measurement variability extends to proprioception assessment as well, where studies utilized different tools including laser pointers and digital inclinometers to measure JPE. These instruments likely yield divergent values due to inherent measurement error differences, making cross-study comparisons challenging. Moreover, current assessment tasks such as the JPE test do not directly measure cervical proprioceptive acuity and may be influenced by confounding factors including body sway, vestibular input, and motor control dysfunction. These inherent limitations may introduce further measurement error and affect the accuracy of proprioceptive evaluation. Methodological quality concerns are further compounded by consistent risk of bias issues identified across all included studies. An important methodological limitation observed across most included studies was the lack of blinding of participants and/or outcome assessors. Given that proprioception outcomes such as JPE and JPS are partially reliant on subjective interpretation and participant cooperation, the absence of blinding can introduce performance and detection biases. Without proper blinding, participants may respond differently due to expectations about the intervention (placebo effects), and assessors may unconsciously influence measurements or data interpretation. This could inflate the perceived effectiveness of the interventions. Only one study attempted a sham control condition to mitigate this bias [19]. Therefore, future trials should incorporate assessor blinding and, where feasible, participant blinding or credible sham controls to improve internal validity. Attrition reporting was similarly problematic, with only three of nine studies adequately documenting dropout rates [20,21,35], while others either reported no attrition or failed to address this critical methodological detail entirely [18,19,31–34]. These reporting gaps make it difficult to assess the true robustness of the findings. Population representation issues present additional constraints on the evidence base's generalizability. The studies exhibited a pronounced age bias, with a mean participant age of 22.6 years that effectively excludes older adults, a population known to experience age-related declines in proprioceptive acuity [44]. This limitation is particularly relevant given that FHP prevalence increases with age and may have different underlying mechanisms in older populations. Similarly, the marked sex imbalance (72% female participants across studies) raises questions about potential sex-specific effects that remain unexplored. No studies addressed possible hormonal influences on proprioception, such as menstrual cycle variations in joint laxity and neuromuscular control that could potentially mediate intervention outcomes in female participants. The GRADE assessment revealed moderate certainty for improvements in JPS and low certainty for JPE, primarily due to methodological heterogeneity, small sample sizes, and inconsistent outcome measures across studies. While the findings support the efficacy of exercise interventions, the limited certainty underscores the need for higher-quality RCTs with standardized protocols and larger, more diverse populations to strengthen clinical recommendations.

## Conclusion

This systematic review demonstrates that current evidence suggests cervical stabilization exercises, the most studied intervention for FHP, may improve cervical proprioception, likely through neuromuscular re-education of deep cervical flexors. While other modalities like cervical retraction show promise, further research is needed to compare efficacy across interventions. To translate these findings into clinical practice, future research should focus on three key areas: (1) establishing standardized diagnostic criteria using specific CVA thresholds for FHP and optimizing exercise prescription, (2) developing optimized intervention protocols incorporating objective adherence monitoring through wearable sensors,

and (3) expanding study populations to include older adults and high-risk occupational groups while stratifying outcomes by age, sex, and comorbidities. Addressing these priorities will enhance intervention reproducibility and improve clinical applicability for diverse populations with FHP-related proprioceptive deficits.

## Supporting information

**S1 File. Prisma 2020-Checklist.**
(DOCX)

## Author contributions

**Conceptualization:** Ali Asghar Norasteh, Kimia Karimi.

**Methodology:** Shabnam Faraji, Fereshteh Ejlali, Mohammad Alghosi, Hamed Zarei.

**Supervision:** Mohammad Alimoradi.

**Writing – original draft:** Ali Asghar Norasteh, Kimia Karimi.

**Writing – review & editing:** Mohammad Alimoradi.

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
