## [Decision Letter · Decision Letter 0]

2 Jul 2025

PONE-D-25-29149Exercise therapy for cervical proprioception in individuals with asymptomatic forward head posture: a systematic reviewPLOS ONE

Dear Dr. Alimoradi,

Thank you for submitting your manuscript to PLOS ONE. After careful consideration, we feel that it has merit but does not fully meet PLOS ONE’s publication criteria as it currently stands. Therefore, we invite you to submit a revised version of the manuscript that addresses the points raised during the review process.

**Kindly make the below changes suggested by the reviewers**

We look forward to receiving your revised manuscript.

Kind regards,

Prateek Srivastav

Academic Editor

PLOS ONE

Journal Requirements:

Reviewers' comments:

Reviewer's Responses to Questions

**Comments to the Author**

1. Is the manuscript technically sound, and do the data support the conclusions?

Reviewer #1: Yes

Reviewer #2: Yes

Reviewer #3: Partly

Reviewer #4: Yes

2. Has the statistical analysis been performed appropriately and rigorously? 

Reviewer #1: Yes

Reviewer #2: Yes

Reviewer #3: Yes

Reviewer #4: Yes

3. Have the authors made all data underlying the findings in their manuscript fully available?

Reviewer #1: Yes

Reviewer #2: Yes

Reviewer #3: Yes

Reviewer #4: Yes

4. Is the manuscript presented in an intelligible fashion and written in standard English?

Reviewer #1: Yes

Reviewer #2: Yes

Reviewer #3: Yes

Reviewer #4: Yes

5. Review Comments to the Author

Reviewer #1: Unclear Definition of 'Diagnostic Inconsistencies': Mentioning "diagnostic inconsistencies" is appropriate, but without a brief example (e.g., varying CVA thresholds), readers are left in the dark.

The phrase “exercise-based interventions” is repeated excessively.

The paper states that exercise may improve proprioception via “neuromuscular re-education” but does not adequately elaborate.

There is no mention of how the lack of blinding in included RCTs may affect bias in outcome measurements.

While ESs are given per direction (flexion, extension), this becomes hard to digest. A table summarizing overall ESs by intervention type (e.g., stabilization vs. WBV) and direction would help.

Statements like “more research is needed” or “future studies should standardize protocols” are expected and lack depth.

Reviewer #2: Suggested improvements:

I do appreciate the motive behind this review manuscript, which is focused on summarizing the current unknowns, and potential benefits of exercise-based interventions in improving cervical proprioception in individuals with forward head posture. This is a well-flowed review article whose theme is suitable to this journal. The objectives are relevant, and the discussion is well-written and flows with the paper. However, the authors overlooked several key and relevant literature that deems this systematic review incomplete and requires major revisions. Mainly since the reviewers did not include “google scholar” as one of the search engines. By running a search using “google scholar”, I found several key RCT publications that could play a major role on the evidence provided in this systematic review, which are provided bellow. The authors also misused several words which portrays a completely different meaning. The manuscript requires major revisions for vocabulary and grammatic errors. Lastly, the discussion could benefit from a subsection that specifically addresses the exercise prescription (i.e., frequency, intensity, type, and duration); specifically, since the authors stated in the introduction that exercise prescription remains unclear and is a clear limitation within the literature. Once these points, and all the other revisions mentioned in the article, are addressed, I will indicate this systematic review research article for publication.

1. Miçooğulları, M., Yüksel, İ., & Angın, S. (2024). Efficacy of scapulothoracic exercises on proprioception and postural stability in cranio-cervico-mandibular malalignment: A randomized, double-blind, controlled trial. Journal of Back and Musculoskeletal Rehabilitation, 37(4), 883-896.

2. Kang, N. Y., Im, S. C., & Kim, K. (2021). Effects of a combination of scapular stabilization and thoracic extension exercises for office workers with forward head posture on the craniovertebral angle, respiration, pain, and disability: A randomized-controlled trial. Turkish journal of physical medicine and rehabilitation, 67(3), 291–299. https://doi.org/10.5606/tftrd.2021.6397

3. Sarıyıldız, A., Deniz, V., & Başaran, S. (2022). Effectiveness of corrective exercise program on alignment, muscle activation and biomechanical properties in forward head posture: a randomized controlled trial. Cukurova Medical Journal, 49(4), 1082-1092.

4. Igbokwe, E. O., Taube, W., & Beinert, K. (2022). A Comparison of the Effects of Stochastic Resonance Therapy, Whole-Body Vibration, and Balance Training on Pain Perception and Sensorimotor Function in Patients With Chronic Nonspecific Neck Pain: Protocol for a Randomized Controlled Trial. JMIR Research Protocols, 11(6), e34430.

5. Gulati, M., Harikesavan, K., & Venkatesan, P. (2025). Immediate effects of thoracic postural correction taping on sensorimotor and respiratory functions in healthy office workers with forward head posture-A randomized controlled trial. Journal of Bodywork and Movement Therapies.

6. Anwar, S., Zahid, J., Alexe, C. I., Ghazi, A., Mareș, G., Sheraz, Z., ... & Gasibat, Q. (2024). Effects of Myofascial Release Technique along with Cognitive Behavior Therapy in University Students with Chronic Neck Pain and Forward Head Posture: A Randomized Clinical Trial. Behavioral Sciences, 14(3), 205.

7. Sarıyıldız, A., Deniz, V., & Başaran, S. (2022). Effectiveness of corrective exercise program on alignment, muscle activation and biomechanical properties in forward head posture: a randomized controlled trial. Cukurova Medical Journal, 49(4), 1082-1092.

8. Dareh-Deh, H. R., Hadadnezhad, M., Letafatkar, A., & Peolsson, A. (2022). Therapeutic routine with respiratory exercises improves posture, muscle activity, and respiratory pattern of patients with neck pain: A randomized controlled trial. Scientific reports, 12(1), 4149.

Specific Recommendations:

See manuscript draft.

Reviewer #3: 1. Summary of the research and my overall impression

Forward head posture (FHP) has various adverse effects, thereby leading to an increasing social demand for its improvement. However, there is no established interventional method, specifically for managing cervical proprioceptive deficits, which is a feature of FHP. As a result, various interventional approaches have been developed. By systematically reviewing existing evidence, this study aimed to clarify the effect of exercise-based interventions on cervical proprioception. This review revealed limitations in this field, such as the lack of established diagnostic criteria and studies with reliable protocols. The authors concluded that although various interventions may be effective in improving cervical proprioception, no definitive conclusions can be drawn due to the low certainty of evidence.

The main strength of this paper is a clear and socially valuable research question. In addition, this systematic review has transparency because this study describes the limitations of each included study in detail. Some of the weaknesses are unclear logical links between the results and the conclusion, especially regarding the comparison among various intervention methods. Therefore, there is a concern that the collected data may not support a part of the conclusion.

Overall, the manuscript is certainly readable and attractive. Especially in the discussion, the possible mechanisms for why interventions improve cervical movements are well-structured. Since this systematic review provides an important contribution to help initiate future reliable studies, I recommend it for publication after some revisions and answers to my questions.

2. Evidence and examples

Major Issues

1. My main concern is related to the claim about the comparison of effectiveness among intervention methods. I will mention three parts: Figure, Discussion, and Conclusion (and Abstract).

• (Figure) As the authors mentioned in the introduction, one of the purposes of this study was to clarify which interventions are most effective among different methods. Therefore, I recommend adding a new figure that summarizes the effect sizes of each intervention method. This visualization would help the readers to interpret the authors’ claims.

• (Discussion) Regarding alternative interventions in the discussion, where it states, “While these alternative modalities …, their relative efficacy compared to conventional stabilization exercises remains unclear (line 342-344 of page 9)”. Indeed, this claim is important to reveal the challenges for the alternative interventions, but the rationale behind this claim is unclear. Please clarify the reason why the effectiveness of alternative intervention methods is unknown compared to traditional interventions for the readers.

• (Conclusion) As pointed out above, the present discussion regarding the comparison of intervention methods seems vague. Therefore, it is unclear which results support a part of the conclusion (“… particularly cervical stabilization exercises, can improve …“, lines 387-388 of page 10). Specifically, a question is: if this conclusion that cervical stabilization exercises were most effective is based on effect size, should the authors consider the fact that the cervical retraction method also had equivalent effect sizes?

(Abstract-result) Regarding the abstract, there is a concern regarding the claim: “Cervical stabilization exercises demonstrated the most consistent benefits, particularly in rotation and flexion tasks” (lines 62-63 of page 2). There are two questions: Firstly, what is this “most” compared with (directions of motion or intervention methods)? Second, if the comparison is regarding intervention methods, what is the logic of this claim? If the claim is based on the number of studies (i.e., the fact that 3/9 are stabilization exercises), it could be misunderstood that alternative intervention methods lacked consistency. To ensure rigor, it is helpful to clarify that consistency regarding alternative interventions could not be evaluated due to the number of studies.

Together, regarding both the conclusion and abstract, the authors might want to consider clarifying the reasons for their claim that stabilization exercises were most or consistently effective. I recommend giving the evidence explicitly in the Discussion. If there is no valid basis, I would recommend that the authors exclude the claim on which interventions are most effective from the conclusions and the result of abstract.

2. Regarding data synthesis and narrative review in the results, the authors grouped the studies that used standard outcome measures as JPE and otherwise as JPS. If this understanding is correct, there is a question about whether it is necessary to separate the included studies into two groups. If the authors interpret that both outcome measures are used to measure joint position sense (i.e., proprioception acuity), the authors could analyze all studies in a unified way without grouping. What are the criteria and rationale for separating JPE and JPS in this analysis?

3. Regarding the conclusion, where it states, "… exercise-based interventions ...can improve cervical proprioception in individuals with forward head posture" (lines 387-389 of page 10). Indeed, the results of this review support the improvement of sensorimotor control using cervical spine movements (i.e., the improvement of “Joint Position Error Test”). However, it appears difficult to identify the mechanism of improvement, as the authors mention it in the Key findings and mechanisms of improvement, too. Therefore, to avoid misleading, the authors might want to consider avoiding a definitive claim that "can improve cervical proprioception."

Minor Issues

1. Page 2 Abstract and Page 4 Search strategy: The authors state “A systematic search of PubMed, Scopus, and Web of Science was conducted up to March 28, 2025” in Abstract-Methods (line 54 of page 2). However, the authors also state, “The search included the electronic databases Web of Science, PubMed, and Scopus, from inception to April 20, 2025…” in Search strategy (lines 132-133 of page 4). Does each statement indicate something different? In addition, did the authors have any restrictions on the publication year of studies they searched?

2. Page 5, lines 163-164 Quality assessment: At the domain of the randomization process, were baseline differences between groups evaluated (if available)?

3. Page 5, Data synthesis: How did the authors treat the results of conditions for comparison (e.g., a result of the SE-VF group in Goo et al. (2024))? Were the results in those conditions excluded because more than one intervention was compared at the same time?

4. Page 5, Data synthesis: For better interpretation for the readers, could the authors fill in item 1 Grouping studies for synthesis, item 3 Describe the synthesis methods, item 7 Data presentation methods, following the SWiM reporting guideline.

5. Page 5, lines 187-189 Data synthesis: Were Kappa statistics conducted between two reviewers at any stage?

6. Page 6, lines 206-207 Study identification and Figure 1: There is a discrepancy between the numbers reported in the text and those reported in Figure 1. Does each indicate something different?

7. Page 6, lines 220-221 Descriptive characteristics of the included studies: Regarding the study referenced as number 30, which did not mention the sex of the participants, which sex group were that participants finally included into?

8. Page 8, line 277 Certainty of the evidence / Page 10, line 380 Limitations and future directions / Table 4: There is a discrepancy between the certainty reported in the text (moderate) and those reported in Table 4 (Low) about JPS. Please correct the discrepancy.

9. Page 8, line 301 Key findings and mechanisms of improvement: This may be a typo: “ES= 1.12=3.66”.

10. Pages 9-10, Limitations and future directions: I suppose that any assessment tools used by the included studies have a limitation that the current measurement tasks (i.e., Joint Position Error Test) cannot evaluate cervical proprioceptive acuity directly. Specifically, there is a possibility that the component of body sway may lead to measurement error, or the fact that it is difficult to isolate the influence of the vestibular system or motor control dysfunction. Are these inherent limitations of assessment tests not mentioned in the section on Limitations and future directions?

11. Figure 2: The black square and black triangle probably represent the mean or median.　Please specify what each symbol indicates and what the error bars demonstrate.

12. References: The cited studies are sometimes misnumbered. Please check the reference list and corresponding numbers throughout the whole text.

Reviewer #4: The manuscript is well structured , evidenced based , methodology aligns well with augment. I therefore endorse the manuscript for publication being very comprehensive structured to provide readers with a clear insight.

6. PLOS authors have the option to publish the peer review history of their article (what does this mean? ). If published, this will include your full peer review and any attached files.

**Do you want your identity to be public for this peer review?** For information about this choice, including consent withdrawal, please see our Privacy Policy .

Reviewer #1: No

Reviewer #2: No

Reviewer #3: No

Reviewer #4: **Yes: ** Ebenezer Ad Adams

---

## [Author Response · Author response to Decision Letter 1]

9 Jul 2025

Dear Reviewer,

Thank you for your valuable feedback on our manuscript titled "Exercise therapy for cervical proprioception in individuals with asymptomatic forward head posture: a systematic review of randomized controlled trials" We have carefully addressed your comments and made significant revisions to improve the quality of our work. Below, you will find our detailed responses to each of your suggestions. We appreciate your insights and look forward to your thoughts on the revised manuscript.

Best regards,

# Reviewer 1

1. Unclear Definition of 'Diagnostic Inconsistencies': Mentioning "diagnostic inconsistencies" is appropriate, but without a brief example (e.g., varying CVA thresholds), readers are left in the dark.

Response: Thank you for your feedback. We have clarified the phrase "diagnostic inconsistencies" by adding an example to improve clarity. Here is the revised text (lines 67 to 69):

“However, methodological heterogeneity and diagnostic inconsistencies—such as variations in craniovertebral angle thresholds used to define FHP—limit the ability to draw definitive conclusions”

2. The phrase “exercise-based interventions” is repeated excessively.

Response: Thank you for your observation. We have revised the manuscript to reduce redundancy by rephrasing and varying the terminology used in place of "exercise-based interventions" where appropriate (e.g., "exercise programs," "training interventions," or simply "interventions").

3. The paper states that exercise may improve proprioception via “neuromuscular re-education” but does not adequately elaborate.

Response: Thank you for your valuable feedback. We have expanded the discussion to elaborate on how neuromuscular re-education contributes to improved proprioception in individuals with FHP. Specifically, we now detail how stabilization exercises retrain deep cervical muscle activation, restore optimal spindle sensitivity, and promote cortical adaptations that enhance proprioceptive integration. Here is the revised text (lines 306 to 314):

“Neuromuscular re-education: Neuromuscular re-education refers to the process of retraining the nervous system to improve the coordination and activation of muscles, particularly those responsible for postural control and joint stability. In the context of FHP, prolonged anterior displacement of the head leads to overactivity of superficial muscles (e.g., sternocleidomastoid and upper trapezius) and underactivity of deep cervical flexors (e.g., longus colli and longus capitis). Stabilization exercises target these deep muscles, promoting their reactivation and restoring optimal muscle activation patterns. This re-education process enhances dynamic joint stability and reduces reliance on superficial muscles, which are prone to fatigue and inefficient for proprioceptive feedback.”

4. There is no mention of how the lack of blinding in included RCTs may affect bias in outcome measurements.

Response: Thank you for your insightful comment. We have now addressed this issue in the “Limitations and future directions” section. Specifically, we discuss how the lack of blinding in most included studies may have introduced performance and detection bias, particularly given the subjective nature of proprioception outcomes. We have also highlighted the need for future studies to implement assessor blinding and credible sham controls to enhance methodological rigor.

Here is the revised text:

“An important methodological limitation observed across most included studies was the lack of blinding of participants and/or outcome assessors. Given that proprioception outcomes such as JPE and JPS are partially reliant on subjective interpretation and participant cooperation, the absence of blinding can introduce performance and detection biases. Without proper blinding, participants may respond differently due to expectations about the intervention (placebo effects), and assessors may unconsciously influence measurements or data interpretation. This could inflate the perceived effectiveness of the interventions. Only one study attempted a sham control condition to mitigate this bias[17]. Therefore, future trials should incorporate assessor blinding and, where feasible, participant blinding or credible sham controls to improve internal validity.”

5. While ESs are given per direction (flexion, extension), this becomes hard to digest. A table summarizing overall ESs by intervention type (e.g., stabilization vs. WBV) and direction would help.

Response: Thank you for your suggestion. To address your comment a new Table was added to manuscript.

Here is the added Table:

Table 4. Summary of effect sizes by intervention type.

Intervention Measurement ES range Magnitude Studies

Cervical stabilization exercise Flexion 0.19-3.66 Trivial to very large Parishan et al. and Goo et al

Joint relocation 1.40 large Lee et al.

Scapular stabilization exercise Joint relocation 1.46 large Lee et al.

Postural correction Extension 0.12 Trivial Jantoon and Uthaikhup

Right rotation 0.86 Moderate

Left rotation 0.63 Moderate

WBV Flexion 0.33 Small Salami et al.

Extension 0.11 Trivial

Backward walking on treadmill Joint relocation 1.19 Moderate Mahmoudi et al.

NASM Flexion 2.14 Very large Abdolahzadeh and Daneshmandi

Extension 0.87 Moderate

Right rotation 2.02 Very large

Left rotation 2.23 Very large

Cervical retraction combined with muscle energy techniques Flexion 3.05-4.55 Very large to nearly perfect Shah et al.

Extension 1.69-4.52 Large to nearly perfect

Right rotation 1.11-3.59 Moderate to very large

Left rotation 1.98-2.97 Large to very large

Conventional physiotherapy Flexion 1.69 Large Panihar and Joshi

Extension 2.48 Very large

Right rotation 3.44 Very large

Left rotation 2.98 Very large

Right flexion 2.30 Very large

Left Flexion 3.13 Very large

Abbreviations: ES, effect size; WBV, whole body vibration; NASM, National Academy of Sports Medicine.

6. Statements like “more research is needed” or “future studies should standardize protocols” are expected and lack depth.

Response: Thank you for your insightful comment. To address your comment, we revised the conclusion of the manuscript. Here is the revised paragraph:

“This systematic review demonstrates that cervical stabilization exercises may lead to improvements in cervical proprioception, likely through neuromuscular re-education of deep cervical flexors. To translate these findings into clinical practice, future research should focus on three key areas: (1) establishing standardized diagnostic criteria using specific CVA thresholds for FHP and optimizing exercise prescription, (2) developing optimized intervention protocols incorporating objective adherence monitoring through wearable sensors, and (3) expanding study populations to include older adults and high-risk occupational groups while stratifying outcomes by age, sex, and comorbidities. Addressing these priorities will enhance intervention reproducibility and improve clinical applicability for diverse populations with FHP-related proprioceptive deficits.”

# Reviewer 2

1. The title should include "systematic review of randomized controlled trials" In addition, provide direction in the title.

Response: Thank you for your comment. The title was revised based on your recommendation.

Here is the revised title:

“Exercise therapy to improve cervical proprioception in individuals with asymptomatic forward head posture: a systematic review of randomized controlled trials”

2. Why did you not include google scholar?

Response: Thank you for your question. The Google Scholar was used as grey literature to identify further pertinent studies. Here is the revised section:

“Additionally, the reference lists of the included studies were screened, and a grey literature search was conducted using Google Scholar to identify further relevant studies, including conference proceedings, dissertations, clinical trial registries, and preprint servers (e.g., medRxiv, bioRxiv).”

3. Can this be the case because it is the most widely used intervention?

Response: Thank you for your valuable feedback. We have revised the text to clarify that cervical stabilization exercises are the most widely studied—rather than definitively the most effective—intervention for FHP, acknowledging limitations in the current evidence. Here is the revised text:

“Current evidence suggests cervical stabilization exercises, the most studied intervention for FHP, may improve cervical proprioception. However, methodological heterogeneity and diagnostic inconsistencies, such as variations in craniovertebral angle thresholds used to define FHP, limit the ability to draw definitive conclusions.”

4. You can also add this citation in here to diversify your references:

Ouattas, A., Wellsandt, E., Hunt, N. H., Boese, C. K., & Knarr, B. A. (2019). Comparing single and multi-joint methods to detect knee joint proprioception deficits post primary unilateral total knee arthroplasty. Clinical Biomechanics, 68, 197-204.

Response: Thank you. It was added.

5. What about threshold to detect passive motion?

Response: Thank you for your comment. The highlighted sentence was revised.

Here is the revised sentence:

“Disruptions in cervical proprioception can lead to impaired joint position sense (JPS), increased joint position error (JPE), and a higher risk of musculoskeletal dysfunction, reduced threshold to detect passive motion, and a higher risk of musculoskeletal dysfunction.”

6. Do you mean decreased angle in here?

Increase in CVA leads to more upward head while decrease in CVA leads to more forward and downward head

Response: Thank you for your comment. It was revised.

Here is the revised sentence:

“One of the most common postural deviations affecting cervical proprioception is forward head posture (FHP), characterized by anterior displacement of the head relative to the shoulders and an decreased craniovertebral angle (CVA)”

7. Reduced not diminished. Diminished portrays almost vanishing.

Response: Thank you. It was revised. Here is the revised sentence:

“The condition is associated with multiple adverse effects, including neck pain, muscle fatigue, reduced respiratory efficiency, and reduced quality of life.”

8. Impaired? The cited papers did not investigate proprioception impairments in clinical populations. Please revise. This should be changed to: "FHP linked to reduced proprioception".

Response: Thank you for your comment. Here is the revised sentence:

“Importantly, FHP has been linked to reduced cervical proprioception, as the altered alignment may disrupt afferent feedback from cervical muscle spindles and joint mechanoreceptors”

9. What is this abbreviation? Please make sure you define abbreviations first.

Response: Thank you. The abbreviation has been mentioned before at line 91.

10. Again, be careful of the word choice. This should be changed to reduced not diminished.

Response: Thank you for your recommendation. Here is the revised sentence:

“Studies suggest that individuals with FHP exhibit greater JPE compared to those with normal posture, indicating reduced proprioceptive acuity.”

11. The discussion could benefit from a subsection that specifically addresses the exercise prescription (i.e., frequency, intensity, type, and duration); specifically, since the authors stated in the introduction that exercise prescription remains unclear and is a clear limitation within the literature.

Response: Thank you for your comment. Here is the new section in the Discussion:

“Analysis of the included studies (Table 3) reveals variability in exercise prescription parameters for improving cervical proprioception in FHP. While cervical stabilization exercises (3-4 sessions/week for 4-6 weeks) emerged as the most consistently effective intervention (ES range: 0.19-3.66), critical gaps remain in our understanding of optimal dosing. Only one study specified exercise intensity (backward walking at 2.4-3.4 km/h), while others lacked objective intensity measures, and none compared different frequencies or durations head-to-head. Alternative approaches like whole-body vibration (single session) and muscle energy techniques (2weeks) showed promise but require longer-term study. Notably, none of the 9 studies compared different frequencies or durations directly, and only 4/9 specified session duration (10–70 minutes). This heterogeneity in frequency (daily to 7×/week), duration (single session to 8 weeks), and unspecified intensities underscores the need for standardized protocols and comparative effectiveness research to establish evidence-based prescription guidelines.”

12. What about Randomized Control Trials? You don't seem to mention it here.

Response: Thank you for your comment. In Table 1, we separated inclusion and exclusion criteria which also included the type of studies. Below you see the Table 1:

Inclusion criteria Exclusion criteria

Population Individuals with FHP without pain

Individuals with ≥ 18 years old

Any sex Individuals with cervical pathology

Intervention Any movement-based intervention More than one intervention was compared at the same time

Comparison Control group without exercise, sham exercise group, or condition for comparison Not applicable

Outcomes Investigate the effect of the intervention on joint position error related to cervical proprioception (e.g. joint position error) No related outcome

Study Design RCTs and non-RCTs Single-group intervention; Case studies; Reviews.

Table 1. Selection criteria for studies.

Abbreviations: RCTs, randomized controlled trials; non-RCTs, non-randomized controlled trials.

13. March 28 or April 20? Make sure you have your dates matching and correct.

Response: Thank you. Here is the revise sentence:

“The search included the electronic databases Web of Science, PubMed, and Scopus, from inception to April 20, 2025, with two authors (KK and MALG) searching independently, with discrepancies resolved through discussion and, if needed, the opinion of a third author.”

I do appreciate the motive behind this review manuscript, which is focused on summarizing the current unknowns, and potential benefits of exercise-based interventions in improving cervical proprioception in individuals with forward head posture. This is a well-flowed review article whose theme is suitable to this journal. The objectives are relevant, and the discussion is well-written and flows with the paper. However, the authors overlooked several key and relevant literature that deems this systematic review incomplete and requires major revisions. Mainly since the reviewers did not include “google scholar” as one of the search engines. By running a search using “google scholar”, I found several key RCT publications that could play a major role on the evidence provided in this systematic review, which are provided bellow. The authors also misused several words which portrays a completely different meaning. The manuscript requires major revisions for vocabulary and grammatic errors. Lastly, the discussion could benefit from a subsection that specifically addresses the exercise prescription (i.e., frequency, intensity, type, and duration); specifically, since the authors stated in the introduction that exercise prescription remains unclear and is a clear limitation within the literature. Once these points, and all the other revisions mentioned in the article, are addressed, I will indicate this systematic review research article for publication.

1. Miçooğulları, M., Yüksel, İ., & Angın, S. (2024). Efficacy of scapulothoracic exercises on proprioception and postural stability in cranio-cervico-mandibular malalignment: A randomized, double-blind, controlled trial. Journal of Back and Musculoskeletal Rehabilitation, 37(4), 883-896.

2. Kang, N. Y., Im, S. C., & Kim, K. (2021). Effects of a combination of scapular stabilization and thoracic extension exercises for office workers with forward head posture on the craniovertebral angle, respiration, pain, and disability: A randomized-controlled trial. Turkish journal of physical medicine and rehabilitation, 67(3), 291–299. https://doi.org/10.5606/tftrd.2021.6397

3. Sarıyıldız, A., Deniz, V., & Başaran, S. (2022). Effectiveness of corrective ex

---

## [Decision Letter · Decision Letter 1]

5 Aug 2025

Exercise therapy to improve cervical proprioception in individuals with asymptomatic forward head posture: a systematic review of randomized controlled trials

PONE-D-25-29149R1

Dear Dr. Alimoradi,

We’re pleased to inform you that your manuscript has been judged scientifically suitable for publication and will be formally accepted for publication once it meets all outstanding technical requirements.

Kind regards,

Prateek Srivastav

Academic Editor

PLOS ONE

Additional Editor Comments (optional):

Reviewers' comments:

Reviewer's Responses to Questions

**Comments to the Author**

1. If the authors have adequately addressed your comments raised in a previous round of review and you feel that this manuscript is now acceptable for publication, you may indicate that here to bypass the “Comments to the Author” section, enter your conflict of interest statement in the “Confidential to Editor” section, and submit your "Accept" recommendation.

Reviewer #2: All comments have been addressed

Reviewer #3: All comments have been addressed

2. Is the manuscript technically sound, and do the data support the conclusions?

Reviewer #2: Yes

Reviewer #3: Yes

3. Has the statistical analysis been performed appropriately and rigorously? 

Reviewer #2: Yes

Reviewer #3: Yes

4. Have the authors made all data underlying the findings in their manuscript fully available?

Reviewer #2: Yes

Reviewer #3: Yes

5. Is the manuscript presented in an intelligible fashion and written in standard English?

Reviewer #2: Yes

Reviewer #3: Yes

6. Review Comments to the Author

Reviewer #2: The authors carefully reviewed and addressed all comments and concerns. I know indicate it for publication.

Reviewer #3: The authors have addressed several questions raised in my previous review. The revised manuscript is well structured, and the data support the revised conclusions. No major concerns remain. However, I encourage the authors to consider the following minor points to improve clarity for readers.

Minor Points

1. Page 6, lines 216, Study identification and Figure 1:

There is a discrepancy in the number of studies reported. The text refers to n = 547, while Figure 1 states n = 574. Clarifying this inconsistency would improve the accuracy of reporting.

2. Page 22, Figure 2:

The black circles appear to indicate the studies with the smallest and largest effect sizes. I recommend revising the figure caption to state this explicitly, which would assist readers in interpreting the figure.

Overall, I recommend this systematic review research article for publication.

7. PLOS authors have the option to publish the peer review history of their article (what does this mean? ). If published, this will include your full peer review and any attached files.

**Do you want your identity to be public for this peer review?** For information about this choice, including consent withdrawal, please see our Privacy Policy .

Reviewer #2: No

Reviewer #3: No

---

## [Editor Report · Acceptance letter]

PONE-D-25-29149R1

PLOS ONE

Dear Dr. Alimoradi,

I'm pleased to inform you that your manuscript has been deemed suitable for publication in PLOS ONE. Congratulations! Your manuscript is now being handed over to our production team.

Kind regards,

on behalf of

Dr. Prateek Srivastav

Academic Editor

PLOS ONE